# A Hybrid Approach to Improve the Video Anomaly Detection Performance of Pixel- and Frame-Based Techniques Using Machine Learning Algorithms

**Hayati Tutar** [1,*] , **Ali Güneş** [1] , **Metin Zontul** [2] **and Zafer Aslan** [1]

1 Department of Computer Engineering, Istanbul Aydin University, Istanbul 34295, Turkey; aligunes@aydin.edu.tr (A.G.); zaferaslan@aydin.edu.tr (Z.A.)
2 Department of Computer Engineering, Sivas University of Science and Technology, Sivas 58000, Turkey; metinzontul@sivas.edu.tr
* Correspondence: hayatitutar@stu.aydin.edu.tr

**Abstract:** With the rapid development in technology in recent years, the use of cameras and the production of video and image data have similarly increased. Therefore, there is a great need to develop and improve video surveillance techniques to their maximum extent, particularly in terms of their speed, performance, and resource utilization. It is challenging to accurately detect anomalies and increase the performance by minimizing false positives, especially in crowded and dynamic areas. Therefore, this study proposes a hybrid video anomaly detection model combining multiple machine learning algorithms with pixel-based video anomaly detection (PBVAD) and frame-based video anomaly detection (FBVAD) models. In the PBVAD model, the motion influence map (MIM) algorithm based on spatio–temporal (ST) factors is used, while in the FBVAD model, the *k*-nearest neighbors (*k*NN) and support vector machine (SVM) machine learning algorithms are used in a hybrid manner. An important result of our study is the high-performance anomaly detection achieved using the proposed hybrid algorithms on the UCF-Crime data set, which contains 128 h of original real-world video data and has not been extensively studied before. The AUC performance metrics obtained using our FBVAD-*k*NN algorithm in experiments were averaged to 98.0%. Meanwhile, the success rates obtained using our PBVAD-MIM algorithm in the experiments were averaged to 80.7%. Our study contributes significantly to the prevention of possible harm by detecting anomalies in video data in a near real-time manner.

**Keywords:** machine learning; computer vision; video surveillance; video anomaly detection; UCF-Crime data set; motion influence map; optical flow; spatio–temporal (ST); *k*-nearest neighbors (*k*NN); support vector machine (SVM)





## 1. Introduction

As a result of the development and widespread use of technology, the rate of production of video and image data has increased rapidly, and the areas of use for cameras, videos, and images are becoming more diverse and widespread every day. The need for image processing in many different fields, such as safety, security, military, intelligence, medicine, road traffic management, urban and public administration, digital agriculture, industrial production, quality control, media, and so on, is constantly growing. Meanwhile, in the near future, with the widespread use of devices that produce and process images instantly (e.g., autonomous land, air, and sea vehicles), the criticality of image processing systems and the need for real-time image and video processing are expected to increase to high levels. Therefore, it is of great importance to develop image-processing techniques and optimize them to the maximum extent, particularly in terms of their speed, performance, and resource utilization [1].

Nevertheless, manual methods based on the human eye are still actively used in many areas of video and image processing and analysis. It is widely known that manual video and image analysis with the human eye takes too long, increases the probability of missing critical images, and entails high workforce and financial costs. On the other hand, considering the rapidly increasing number and duration of videos, it is understood that relying on manual analysis methods using the human eye will become more and more difficult, and will ultimately be unable to meet the associated demands [2].

The main objective of this study is video anomaly detection. It is not always possible to watch video recordings from beginning to end when seeking to detect unusual (i.e., anomalous) situations, especially when considering long-duration videos. Therefore, it is of great importance to classify video frames by analyzing them with machine learning techniques. After the analysis and classification of video frames, the aim is to automatically detect anomalous frames in a near real-time manner [3], allowing the operator (or another concerned party) to take action by focusing on these images. In this way, the possibility of overlooking unusual situations can be minimized, anomalous situations will be detected in near real time, and their possible harmful effects may be prevented. As such, significant contributions can be made to human health and public safety. At the same time, this study aims to provide many indirect contributions, such as those related to environmental protection and the efficient use of time, facilities, and resources.

Our study proposes a hybrid video anomaly detection model combining multiple machine learning algorithms with pixel-based video anomaly detection (PBVAD) and frame-based video anomaly detection (FBVAD) models. In the PBVAD model, the motion influence map (MIM) algorithm based on spatial–temporal (ST) elements [4] is used, while in the FBVAD model, the *k*-nearest neighbors (*k*NN) and support vector machine (SVM) machine learning algorithms are used in a hybrid manner. First, video frames are analyzed with machine learning techniques. Then, focusing on the obtained data, the anomalous images are detected again with machine learning techniques. After this, the classification of the detected anomalies is performed.

UCSD Ped1 [5] and UCSD Ped2 [6] are the most commonly used data sets in relevant studies, and the anomaly detection success rates yielded in recent studies based on these data sets have reached 97% and above (UCSD Ped1 [7], UCSD Ped2 [8]). Meanwhile, the average success rates using other data sets are lower. In particular, it has been observed that the success rates in studies conducted on the UCF-Crime data set were lower than those conducted on other data sets [9,10]. Therefore, in our study, we focus in particular on the UCF-Crime data set.

In the implementation and testing phases of our work, we focused on video anomaly detection in relation to important incidents that are closely related to societal and public safety, such as abuse, assault, burglary, fighting, explosions, and road accidents, as contained in the UCF-Crime data set.

For our study, a comprehensive literature review was first conducted. Within the scope of the literature review, the data sets, feature extraction techniques, developed models, algorithms, development environments, analysis results, and success rates reported in previous studies are examined in detail. Then, the used methodology and technical methods are presented. The subsequent part of the study describes the developed application and our test procedures, while the final section provides an evaluation of the findings and test results obtained in our study.

In this study, the methodologies, algorithms, models, and anomaly detection techniques used in various video anomaly detection studies are analyzed, and their performances are evaluated comparatively.

## 2. Literature Review

We conducted a literature review focused on the most recent studies published after 2018, in relation to our research area.

One of our primary findings is that image analysis studies have been carried out in various areas of interest on a subject-by-subject basis but have not been conducted to offer references for many general fields [11–13]. Some studies have emphasized the need for real-time image processing [14]. However, the number of studies comparing multiple machine learning techniques is limited [14].

During the literature review process, it was observed that studies on video anomaly detection—which is our area of interest—have increased in number, especially in recent years [15].

### 2.1. Using Methods and Algorithm Types

When the video anomaly detection studies were examined methodologically, we observed that different methodologies were used in each, including spatio–temporal I3D AE, temporal segment C3D, convolutional LSTM, deep learning (CNN/RNN), U-Net, two U-Net blocks, LSTM network, bidirectional LSTM prediction (BD-LSTM), dual discriminator GAN, and multi-timescale [15] methods. For example, Gianchandani et al. have used the spatio–temporal I3D AE anomaly detection method [16], Sultani et al. have used the temporal segment C3D anomaly detection method [9], Ullah et al. have used the bidirectional LSTM anomaly detection method [17], and Wu et al. have used deep learning (CNN/RNN) algorithm-based anomaly detection methods [18].

By examining the techniques used in the literature, it can be observed that anomaly detection techniques such as sparse coding [6,19,20], weakly supervised learning [21,22], spatio–temporal [4,23], normality learning-based [24–27], Gaussian mixture [28–30], graph-based [31,32], autoencoder technique [33–35], unsupervised anomaly detection [36,37], self-supervised learning [38–41] and probabilistic [42,43] models have been used.

In addition to the algorithms and learning methods used, it has been observed that whether or not a feature extraction technique is used for frame- or pixel-based analysis may also affect the success of anomaly detection [10].

Georgescu et al. [39] have studied anomaly detection in videos through self-supervised and multi-task learning using the 3D CNN and YOLOv3 algorithms with the MS COCO and ResNet-50 methodology. In particular, they approached anomalous event detection in videos through self-supervised and multi-task learning at the object level. They first utilized a pretrained detector to detect objects, then trained a 3D convolutional neural network to produce discriminative anomaly-specific information by jointly learning multiple proxy tasks: three self-supervised and one based on knowledge distillation. In their experimental results, they stated that their lightweight architecture outperformed state-of-the-art methods when applied to three benchmark data sets: Avenue, ShanghaiTech, and UCSD Ped2 [39].

Hao et al. [44] have studied effective crowd anomaly detection through spatio–temporal texture analyses using Gabor-filtered textures and SVM algorithms and described two major breakthroughs. First, their developed spatio–temporal texture extraction algorithm was able to effectively extract textures from videos with an abundance of crowd motion details, which was achieved by adopting Gabor-filtered textures with the highest information entropy values. Second, they devised a novel scheme for defining crowd motion patterns (signatures), in order to identify abnormal behaviors in the crowd, by employing an enhanced gray level co-occurrence matrix model. In their experiments, various classic classifiers were utilized to benchmark the performance of the proposed method [44].

Lee V.T. et al. [45] have studied attention-based residual auto-encoders for video anomaly detection using deep (3D) CNN and ConvLSTM algorithms. They proposed a system that adopts a spatial branch and temporal branch in a unified network, allowing for the effective exploitation of both spatial and temporal information. Their experimental results revealed AUC performance rates of 97.4% for UCSD Ped2, 86.7% for CUHK Avenue, and 73.6% for ShanghaiTech [45].

In some studies, anomaly detection in crowded scenes has been undertaken using convolutional neural network (CNN)-based algorithms. For example, Ravanbakhsh et al. [46]

have published the study "plug-and-play CNN for crowd motion analysis: an application in abnormal event detection," in which they sought to measure local abnormalities by combining the semantic information inherited from existing CNN models with low-level optical flow [46]. Sabokrou et al. [47] have studied deep and fast anomaly detection in crowded scenes using fully convolutional neural networks (FCNs) and temporal data; here, a pretrained supervised FCN was transferred to an unsupervised FCN, ensuring the detection of (global) anomalies in scenes [47]. As another example, Smeureanu et al. [48] have studied the application of real-time deep learning methods for abandoned luggage detection video recognition, based on a cascade of convolutional neural networks (CNNs) [48].

### 2.2. Using Data set Types

In the video analysis and anomaly detection studies outlined in the literature, we found that the UCSD Ped1 [5], UCSD Ped2 [49], subway entrance–exit [43], UMN [50], Avenue [51], UCF-Crime [9], ShanghaiTech [52], and XD-Violence [53] data sets were commonly used.

Sultani et al. [9] have studied human behavior and the automatic detection of abnormal events by creating a 128 h data set (UCF-Crime) containing original video recordings of the real world [9]. They sought to identify anomalies using a deep multiple-instance ranking framework by leveraging weakly labeled training videos, where the training labels (i.e., anomalous or normal) were set at the video level, instead of the clip level. In their approach, they considered normal and anomalous videos as bags and video segments as instances in multiple-instance learning (MIL), facilitating the automatic learning of a deep anomaly ranking model that predicted high anomaly scores for anomalous video segments [9].

Xia et al. [54] have studied video abnormal event detection using SVM, deep neural network (DNN), CNN, generative adversarial network (GAN), and one-class neural network (ONN) algorithms. In their experimental results, they achieved AUC rates of 94.9% on the PED1 data set and 94.5% on the PED2 data set [54].

In some studies, method- and data set-based performance comparisons in relation to anomaly event detection have been addressed, and the success rates of the methods used typically differ depending on the data set [53,55].

### 2.3. Sample Video Anomaly Detection Techniques and Application Types

Anomaly detection in video frames can be used in many different sectors. For example, studies on anomaly detection in crowded areas have recently been carried out [56].

Some of the studies focused on event-based anomaly detection are also examined here, and by evaluating their result graphs, it was observed that these studies detected issues such as fighting, abuse, and shootings with a higher success rate, especially in the context of social events, while their success rates were lower in relation to detecting events such as explosions and road accidents [10]. It remains to be investigated whether the inclusion of human behavior-based learning data in the data sets used is also effective.

Liu et al. [3] have developed an expert real-time system for anomaly detection in aerators based on computer vision technology and existing surveillance cameras. They proposed a novel algorithm called the "Reference Frame Kanade–Lucas–Tomasi (RF-KLT)" algorithm for motion feature extraction in fixed regions. The proposed expert system performed real-time, robust, and cost-free anomaly detection in aerators using both the actual video data set and an augmented video data set [3].

In some studies, spatio–temporal-based algorithms have been used as a reference. Bertini et al. [57] have proposed an approach for anomaly detection and localization in video surveillance applications based on spatio–temporal features, through capturing a scene's dynamic statistics together with its appearance [57]. In another example, Jiang et al. [58] have studied anomalous video event detection using the spatio–temporal context, and proposed a context-aware method to detect anomalies in which three different spatio–temporal context levels are considered. In their experiments on real traffic videos, they identified video anomalies that are hazardous or illegal according to relevant traffic

regulations [58]. As another example, Li N. et al. [59] have conducted spatio–temporal context analyses within videos for anomalous event detection and localization. Their approach employs an unsupervised statistical learning framework based on the analysis of spatio–temporal video volume configurations within video cubes, which learns global activity patterns and local salient behavior patterns through clustering and sparse coding, respectively [59]. In the last example, Li W. et al. [30] have proposed a method for the detection and localization of anomalous behaviors in crowded scenes and developed a joint detector of temporal and spatial anomalies [5].

Wang et al. [60] have studied real-time and accurate object detection using an approach that compresses videos through long short-term feature aggregation. They proposed a novel short-term feature aggregation method to propagate the rich information present in key frame features to non-key frame features in a rapid manner. Their experiments on a large-scale ImageNet VID benchmark achieved a 77.2% mAP (which is on par with state-of-the-art accuracy) at a speed of 30 FPS using a Titan X GPU [60].

In some studies, GAN algorithms have been used as a reference. As an example, Jiang et al. [61] have proposed a background-agnostic framework that learns from training videos containing only normal events. Their framework is composed of an object detector, a set of appearance and motion auto-encoders, and a set of classifiers [61]. In another example, Ravanbakhsh et al. [62] have used generative adversarial networks (GANs), which were trained using normal frames and the corresponding optical flow images, in order to develop an internal representation of a scene's normality [62].

Zaheer et al. [63] have studied clustering-assisted weakly supervised learning with normalcy suppression for anomalous event detection using the UCF-Crime and ShanghaiTech data sets. They proposed a weakly supervised anomaly detection method that offers three-stage contributions. In their experimental results, they achieved an 83.03% and 89.67% frame-level AUC on the UCF-Crime and ShanghaiTech data sets, respectively [63].

In other studies, research has been conducted on improving video anomaly detection algorithms. Sikdar et al. [64] have developed an adaptive training-less framework for anomaly detection in crowd scenes. Their proposed solution comprises a pipeline consisting of three major components: namely, an adaptive 3D-DCT model for multiobject detection-based association, local motion descriptor generation through an improved-saliency guided optical flow, and anomaly detection based on the Earth mover's distance (EMD) [64]. Tudor I. et al. [65], in an attempt to unmask the abnormal events in videos, have proposed a novel framework for abnormal event detection in videos that requires no training sequences [65]. Xu D. et al. [66] have attempted to detect anomalous events in videos by learning deep representations of appearance and motion, and proposed Appearance and Motion DeepNet (AMDN), a novel approach based on deep neural networks that is intended to automatically learn feature representations. In addition, based on the learned features, multiple one-class SVM models were used to predict the anomaly scores of each input. Finally, a novel late fusion strategy was proposed to combine the computed scores and detect abnormal events [66].

Sarikan et al. [67] have studied the detection of anomalies in vehicle traffic using image processing and machine learning algorithms. In another study, performance comparisons of anomaly event detection at the energy consumption level have been discussed, and the results were evaluated [68].

Kiranyaz et al. [69] have studied real-time phonocardiogram (PCG) anomaly detection. They deliberately focused on the anomaly detection problem, assuming a reasonably high signal-to-noise ratio (SNR) in the records. By using 1D convolutional neural networks trained with a novel data purification approach, they aimed to achieve the highest detection performance and real-time processing ability with a significantly lower delay and computational complexity. In their experimental findings, they stated that further improvements will require a personalized (i.e., patient-specific) approach to avoid the major drawbacks of a global phonocardiogram (PCG) classification approach [69].

A recent study has used quantum technologies and hybrid machine learning algorithms to analyze video frames in the healthcare domain [70]. In comparing the use of the hybrid quantum CNN (HQCNN) algorithm with the classical CNN machine learning algorithm, it was found that the HQCNN technique performed better in analyzing and diagnosing medical images [70].

Other than these, different methods have been used for human activity recognition [71,72], video object detection [73–75], video motion anomaly detection [76–79], online anomaly detection [80,81], video anomaly detection by injecting temporal information in feature extraction [82], anomaly detection method with deep support vector data description (DSVDD) using deep learning algorithm [83], video anomaly detection method with a main-auxiliary aggregation strategy (MAAS) [84], and the analysis of anomalies with feature embeddings of pre-trained CNNs with the use of novel cross-domain generalization measures [85] in various studies.

## 3. Methodology

In this section, the architecture, algorithms, and methodologies used for the development of our proposed model are discussed.

### 3.1. Modeling and Algorithm Selection

Anomaly detection in crowded areas is a very challenging task. In particular, minimizing false positives and achieving a high detection performance is challenging. For this reason, we propose a hybrid anomaly detection model by combining multiple algorithms and techniques.

Regarding the choice of algorithm in the proposed model, both supervised and unsupervised learning algorithms are selected, thus enabling a hybrid approach. In anomaly detection studies in particular, the success rates of unsupervised learning algorithms are very high when applied for the detection of unusual situations and events that have not been previously defined. However, when seeking to prevent or minimize false positives, supervised learning algorithms provide significant advantages. The support vector machine (SVM) and $k$-nearest neighbors ($k$NN) algorithms were found to be preferred as supervised learning algorithms in the reference literature. In addition, the motion influence map (MIM) algorithm, which is successful in detecting human behavior and is based on spatio–temporal (ST) features, is also used effectively in our study [86].

The most important phase of video analysis and anomaly detection is the feature extraction process, which is also the longest part of the whole process. After the feature extraction, video frames are sent through both the supervised and unsupervised algorithm processes separately and are analyzed simultaneously. After this analysis, anomaly detection is performed, and a scoring (rating) and labeling process is performed on the detected anomalies. Video frames that are detected as common anomalies in both processes are labeled as high-grade anomalies.

An architectural diagram of our model is shown in Figure 1. The structure and components of our proposed model are described in the subsequent sections.

### 3.2. Model Architecture Diagram

The architecture of the proposed model basically consists of three layers. The layers of this architecture are as follows (see Figure 1 for an architectural diagram of the model):

(a)   *Input layer;*
(b)   *Video surveillance and anomaly detection process layer;*
(c)   *Output layer.*

We briefly summarize the layers and components of the architectural diagram in the following subsections:

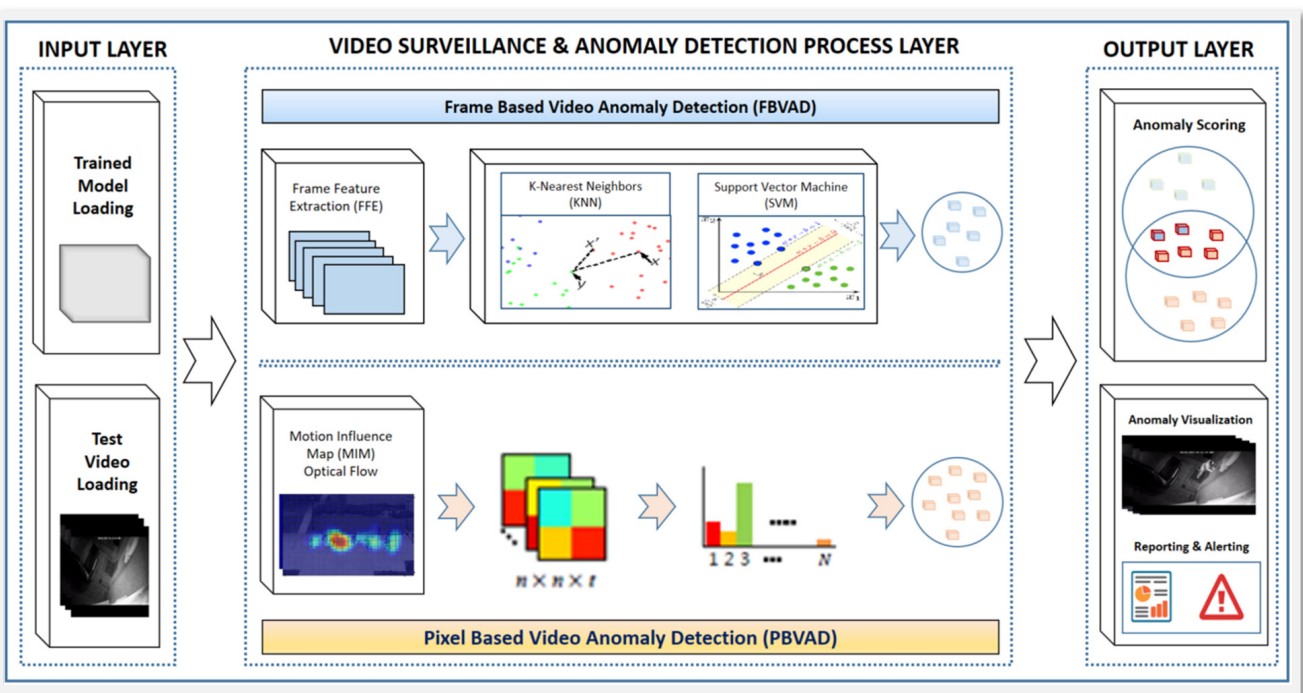

**Figure 1.** Architectural diagram of proposed anomaly detection model using hybrid machine learning algorithms.

### 3.2.1. Input Layer

This layer contains the components to be entered into the analysis process and consists of "Trained Model Data" and "Test Video Data" components. Operations are initiated in this layer.

### 3.2.2. Video Surveillance and Anomaly Detection Process Layer

This layer is where the main components of the analysis process are located, in which anomaly detection is performed. It consists of two main components:

-   Anomaly detection with frame-based ($k$NN, SVM) algorithms;
-   Anomaly detection with pixel-based (motion influence map; MIM) algorithms.

### 3.2.3. Output Layer

This is the layer in which the anomalous data and images detected as a result of the analysis process are displayed. This layer is interactive and comprises anomaly visualization and anomaly reporting and alert components.

### 3.3. *Frame-Based Video Anomaly Detection (FBVAD) Method*

For frame-based anomaly detection based on video frames, machine learning algorithms including $k$-nearest neighbors ($k$NN) and support vector machine (SVM) are used. These algorithms are briefly described in the following subsections:

### 3.3.1. Frame-Based Feature Extraction

Feature extraction is one of the most important aspects of video analysis and anomaly detection, directly affecting the accuracy of the result. The feature extraction process, as shown in Figure 2, consists of the following steps:

*(i)    Obtaining Frames*

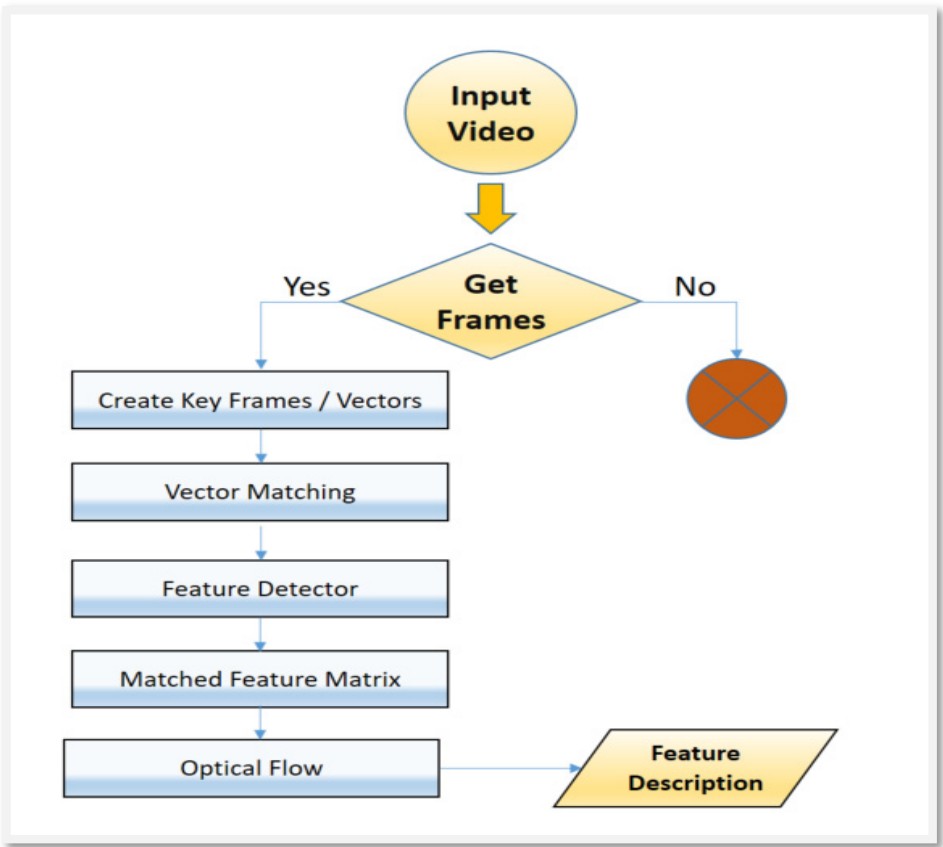

**Figure 2.** Frame-based video feature extraction phases.

In the feature extraction process, when the test video data are loaded, they are first checked for validity, and it is ascertained whether the frames can be opened. If the frames cannot be opened, the process is terminated with a warning message. If the frames can be opened, the content extraction process steps are commenced and progress one by one for each frame in a loop, passing from the first to last frame of the video data. New and existing frames are compared.

*(ii)* Creating Key Frames/Vectors

In this phase of the feature extraction process, key vectors in the frame image are detected, and the descriptors in the key vector position image are calculated.

*(iii)* Matching Vectors

In this phase of the feature extraction process, matches are made between the vectors in the frame image and the key vectors.

*(iv)* Detecting Features

In this phase of the feature extraction process, the features in the frame are detected.

*(v)* Feature Matching Matrix

In this phase of the feature extraction process, the detected features and vectors are located by matching them in relation to the key frames/vectors. The $k$NN algorithm is used in this process, for which the value of $k$ is set to 2. Additionally, the uniqueness threshold is set to 0.80.

*(vi)* Optical Flow

In this phase of feature extraction, the optical flow is calculated for a sparse feature set using the iterative Lucas–Kanade method. The "minEigThreshold" value is set to 0.0001 in the procedure.

*(vii)* Feature Description

In this phase of the feature extraction process, all the results are returned as part of array sets of processed data to the main processing point, from which the process began.

### 3.3.2. Video Anomaly Detection with k-Nearest Neighbors (kNN) Algorithm

The *k*-nearest neighbors (*k*NN) model is used to classify, train, and save the data. The training process takes the training and test data and their labels as parameters. The value of the *k* value is set to 5 for training. Anomaly detection, classification, and scoring are performed using frame-based feature extraction values.

The Algorithm 1 *k*-nearest neighbors (*k*NN) is defined below.

---

**Algorithm 1 *k*-Nearest Neighbors (*k*NN)**

---

1. $k^* = \left\lfloor Bn \frac{4}{d+4} \right\rfloor$
2. $w^*{}_{ni} = \frac{1}{k^*} \left[ 1 + \frac{1}{2} - \frac{d}{2k^{*2/d}} \left\{ i^{1+2/d} - (i-1)^{1+2/d} \right\} \right]$
3. $for\ \boldsymbol{i} = 1, 2, \ldots \boldsymbol{k}^*\ and$
4. $w^*{}_{ni} = 0$
5. $for\ \boldsymbol{i} = \boldsymbol{k}^* + 1,\ \ldots, n$

---

### 3.3.3. Video Anomaly Detection with Support Vector Machine (SVM) Algorithm

The data are classified using the support vector machine (SVM) classifier, and the model is saved. The training process takes the training and test data and their labels as parameters. Anomaly detection, classification, and scoring are performed using frame-based feature extraction values.

### *3.4. Pixel-Based Video Anomaly Detection (PBVAD) Method*

The motion influence map (MIM) algorithm is used for pixel-based anomaly detection in video frames. The general architectural framework of the motion influence map (MIM) algorithm is described below.

### 3.4.1. Motion Influence Map (MIM) Algorithm

The motion influence map (MIM) algorithm basically consists of four steps, which are listed below:

*(i)* Optical flow;
*(ii)* Calculation of the effect of movement between blocks;
*(iii)* Calculation of impact weights between both blocks;
*(iv)* Calculation of motion ray (direction) weights for each block.

A diagram of pixel-based feature extraction using the motion influence map (MIM) algorithm is shown in Figure 3 below [86].

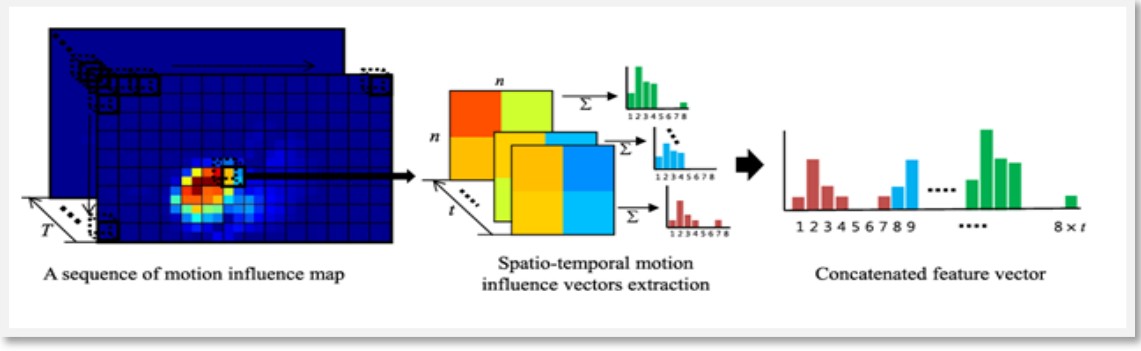

**Figure 3.** Motion influence map pixel-based feature extraction diagram [27].

The Algorithm 2 motion influence map (MIM) is defined below.

---

**Algorithm 2 Motion Influence Map (MIM)**

---

$\textbf{\textit{INPUT}} : MV \leftarrow motion\ vector\ set, \ BS \leftarrow block\ size\ x\ y, \ CB \leftarrow a\ centri\ of\ blocks$
$\textbf{\textit{OUTPUT}} : MIP \leftarrow motion\ influence\ map$
 $\textbf{\textit{for}}\ i\ \in BS(x)\ \textbf{\textit{do}}$
  $\textbf{\textit{for}}\ j\ \in BS(y)\ \textbf{\textit{do}}$
    $b_i = \frac{1}{j}\sum_j OF_i^j$
    $T_d = \parallel b_i \parallel \times BS$
    $OF_k = \parallel MV_{ij1} \parallel$
    $+TAF_i = \angle OF_k + \frac{\pi}{2}$
    $-TAF_i = \angle OF_k - \frac{\pi}{2}$
   $\textbf{\textit{for}}\ p\ \in BS(x)\ \textbf{\textit{do}}$
   $\textbf{\textit{for}}\ q\ \in BS(y)\ \textbf{\textit{do}}$
      $Calculate\ the\ Euclidean\ Distance\ ED(i,j,p,q)\ between\ CB(i,j,p,q)$
      $\textbf{\textit{if}}\ E(i,j,p,q) < T_d\ \textbf{\textit{then}}$
        $Calculate\ the\ Angle\ A_{ij}\ between\ b_i\ and\ b_j$
        $\textbf{\textit{if}}\ +TAF_i < A_{ij} < -TAF_i\ \textbf{\textit{then}}$
          $MIP^{pq}\left(\angle b_i\right) = MIP^{pq}\left(\angle b_i + \exp\left(\frac{ED_{(i,j,p,q)}}{\parallel MV_{ij0}\parallel}\right)\right)$
        $\textbf{\textit{end if}}$
      $\textbf{\textit{end if}}$
     $\textbf{\textit{end for}}//(q)$
     $\textbf{\textit{end for}}//(p)$
  $\textbf{\textit{end for}}//(j)$
 $\textbf{\textit{end for}}//(i)$

---

### 3.4.2. Pixel-Based Video Anomaly Detection with Motion Influence Map (MIM) Algorithm

The motion influence map (MIM) algorithm is used for pixel-based anomaly detection in video frames. The general architectural framework of the motion influence map (MIM) algorithm is depicted in Figure 4 below.

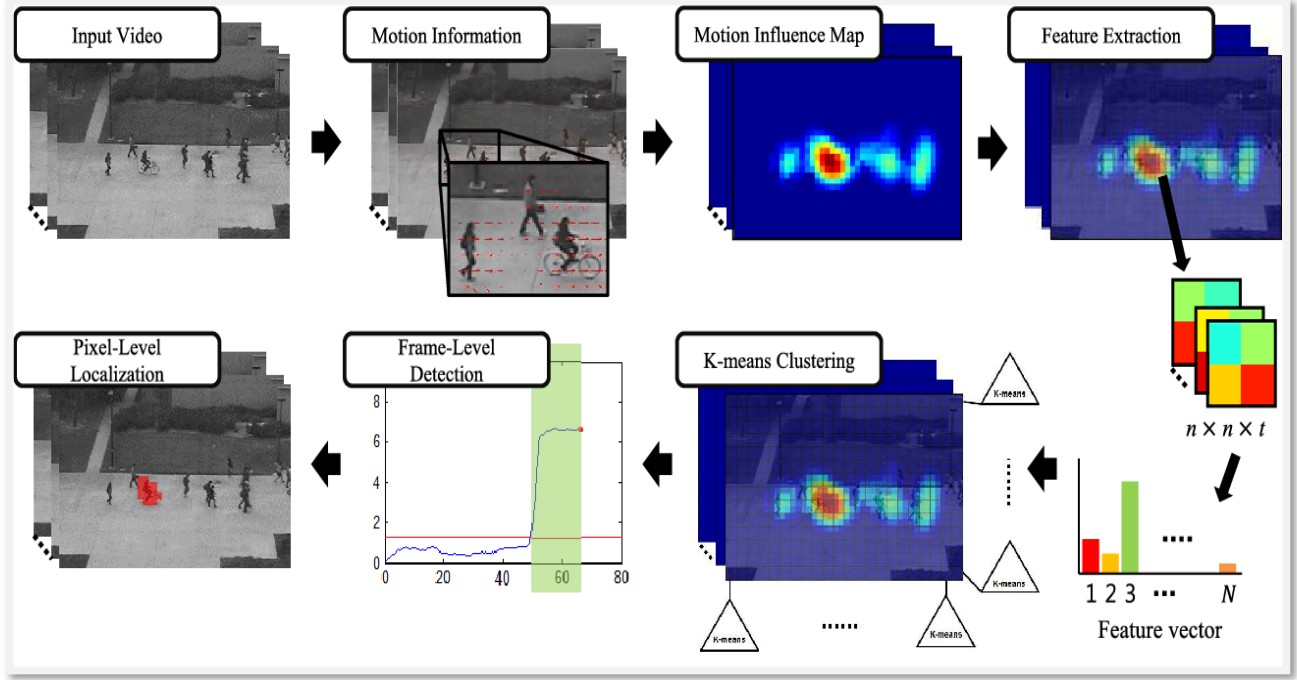

**Figure 4.** Anomaly detection diagram using the motion influence map (MIM) [86].

Motion information is computed at the pixel and cluster levels by inputting a sequence of video frames. The kinetic energy of each block is then calculated to structure the motion influence map. After creating a motion influence map of the frame (scene), we can identify the set containing an event suspected of having a characteristic motion vector. This event is tracked through several consecutive scenes, such that we can extract the feature bundle for each adjacent block through a megablock containing a certain number of scenes. Thus, each frame will be divided into a group of megablocks, each containing the motion influence information. Finally, by summing up the vectors of motion influence in each frame separately, we extract the temporal and spatial properties of several megablocks. In order to not classify normal events as suspicious, anomalous events are separated. To identify the centers of normal events in the image frames, the *k*-means algorithm is first applied. Then, the Euclidean distance law is applied to the monitored image frame and the centers of other image frames to detect the anomalous event. If the event exceeds a certain threshold, it is recognized as an anomalous event scene.

## 4. Experiments and Results

This section discusses the implementation and test results in detail. In addition, the data sets used for the implementation of the proposed application, evaluation criteria, test results, and detected anomalies are assessed.

### 4.1. General Information on Implementation and Test Environment

The test application was developed in Windows Visual Studio using the C# programming language. A C#-compatible version of the OpenCV library was used for the video analysis and computer vision processing. In addition, the Python programming language was used for some of the development and testing steps. For the test environment, we used a computer with Windows 10 64-bit operating system, i7 and 4 dual-core CPUs, an 8-core 2.8 GHz Intel processor, and 32 GB RAM.

### 4.2. Data Set Selection and Implementation

Most recent studies on common data sets, such as UCSD Ped1 [87] and UCSD Ped2 [8], have yielded success rates of 97.1% and above. However, as the success rates reported in studies utilizing the UCF-Crime [9] data set are generally lower than those for other data sets [10], the UCF-Crime data set was preferentially used in our research and testing process.

The UCF-Crime data set is a large-scale anomaly detection data set containing 1900 unprocessed videos captured by road, street, and indoor surveillance cameras in the real world, with a total duration of 128 h [9]. Unlike the static backgrounds in the ShanghaiTech data set, UCF-Crime features complex and diverse backgrounds. The training and test sets contain the same numbers of normal and abnormal videos. The data set covers 13 different anomaly classes through 1610 training videos with video-level labels and 290 test videos with frame-level labels.

In the implementation process, the algorithms to be used were first determined. Necessary installations and configurations were developed for the algorithms in the development environment. The data sets were then made available in the test environment, and experimental studies were carried out.

The reference data for the UCF-Crime data set used in our test study are presented (by category) in Table 1 below.

**Table 1.** Reference information for UCF-Crime data set used in our study.

| Data Set | File | Frame | Video Length |
|---|---|---|---|
| Abuse | Abuse002_x264.mp4 | 865 | 00:28 s |
| Assault | Assault002_x264.mp4 | 2523 | 01:24 s |
| Burglary | Burglary012_x264.mp4 | 1698 | 00:56 s |
| Explosion | Explosion045_x264.mp4 | 757 | 00:25 s |
| Fighting | Fighting006_x264.mp4 | 944 | 00:31 s |
| Road accidents | Road Accidents002_x264.mp4 | 347 | 00:11 s |

*4.3. Test Results and Anomaly Detection Evaluation*

As mentioned above, the UCF-Crime data set was taken as a reference in our study, and anomaly detection tests were carried out on subdata sets concerning the categories of abuse, assault, burglary, explosion, fighting, and road accidents. In our study, image clarity was the main issue directly affecting the anomaly detection success rates of the algorithms, and the anomaly detection performance was quite low when using camera data with unclear, low quality images. On the other hand, the distance and proximity of the camera to the crime scene were found to be the main factors that directly affected the performance of the anomaly detection algorithms.

The performance results on the UCF-Crime data set, regarding pixel-based video anomaly detection (PBVAD) using the MIM algorithm and frame-based video anomaly detection (FBVAD) using the *k*NN and SVM algorithms, are shown in Table 2 and Figure 5 below. In the table and graph, the AUC results for each algorithm are shown separately, based on the considered categories.

**Table 2.** Video anomaly detection AUC rate results for the tested algorithms.

| Data Set | FBVAD-*k*NN AUC (%) | FBVAD-SVM AUC (%) | PBVAD-MIM AUC (%) |
|---|---|---|---|
| Abuse | 98.80 | 77.60 | 74.20 |
| Assault | 97.90 | 75.13 | 86.60 |
| Burglary | 96.60 | 75.11 | 75.80 |
| Explosion | 99.30 | 78.70 | 73.10 |
| Fighting | 96.80 | 77.20 | 78.80 |
| Road accidents | 98.30 | 80.10 | 95.80 |

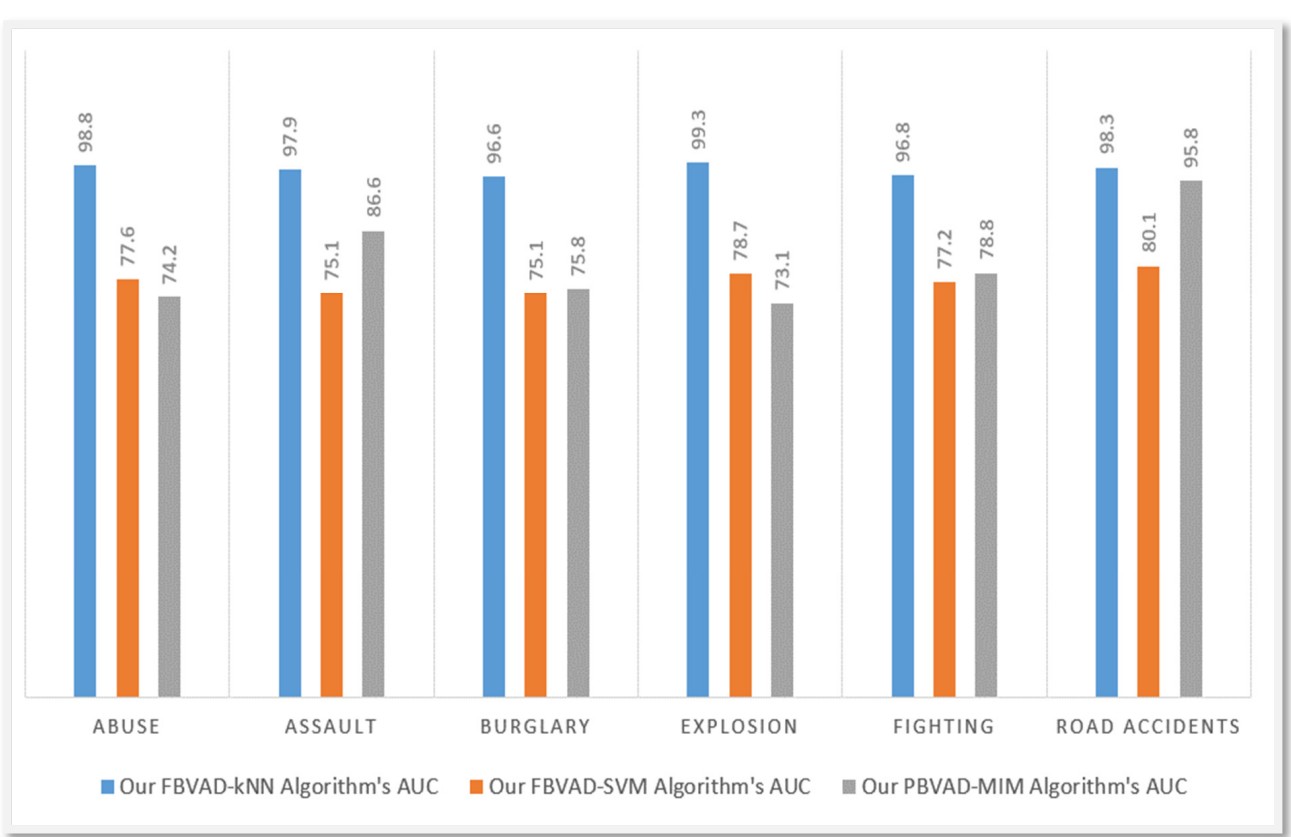

**Figure 5.** Frame-based (FBVAD) and pixel-based (PBVAD) video anomaly detection AUC rates on the UCF-Crime data set.

The AUC performance and score results on the UCF-Crime data set for pixel-based video anomaly detection (PBVAD) using the MIM algorithm and frame-based video anomaly detection (FBVAD) using the *k*NN and SVM algorithms are shown in Tables 3–5 below, which list the AUC, precision, sensitivity, and F-Score values for each of the categories in the UCF-Crime data set.

**Table 3.** Video anomaly detection AUC rate and results for FBVAD-*k*NN algorithm.

| Data Set | AUC | Precision | Sensitivity | F-Score |
|---|---|---|---|---|
| Abuse | 98.80 | 0.998 | 0.9762 | 0.974 |
| Assault | 97.90 | 0.996 | 0.9569 | 0.955 |
| Burglary | 96.60 | 0.993 | 0.9325 | 0.928 |
| Explosion | 99.30 | 0.999 | 0.9875 | 0.984 |
| Fighting | 96.80 | 0.994 | 0.9372 | 0.925 |
| Road accidents | 98.30 | 0.998 | 0.9565 | 0.955 |

The AUC performance and score results for FBVAD using the SVM algorithm, based on the UCF-Crime data set, are shown in Table 4 below.

**Table 4.** Video anomaly detection AUC rate and results for FBVAD-SVM algorithm.

| Data Set | AUC | Precision | Sensitivity | F Score |
|---|---|---|---|---|
| Abuse | 77.60 | 0.998 | 0.7757 | 0.874 |
| Assault | 75.13 | 0.992 | 0.7515 | 0.857 |
| Burglary | 75.11 | 0.992 | 0.7520 | 0.858 |
| Explosion | 78.70 | 0.995 | 0.7873 | 0.881 |
| Fighting | 77.20 | 0.994 | 0.7722 | 0.871 |
| Road accidents | 80.10 | 0.996 | 0.8012 | 0.890 |

The AUC performance and score results for PBVAD using the MIM algorithm, based on the UCF-Crime data set, are shown in Table 5 below.

**Table 5.** Video anomaly detection AUC rate and results for PBVAD-MIM algorithm.

| Data Set | AUC | Precision | Sensitivity | F Score |
|---|---|---|---|---|
| Abuse | 74.20 | 0.484 | 0.7419 | 0.652 |
| Assault | 86.60 | 0.731 | 0.8657 | 0.845 |
| Burglary | 75.80 | 0.516 | 0.7581 | 0.681 |
| Explosion | 73.10 | 0.462 | 0.7308 | 0.632 |
| Fighting | 78.80 | 0.576 | 0.7879 | 0.731 |
| Road accidents | 95.80 | 0.917 | 0.9583 | 0.957 |

The confusion matrix reference and prediction values for pixel-based video anomaly detection (PBVAD) using the MIM algorithm and frame-based video anomaly detection (FBVAD) using the *k*NN and SVM algorithms, based on the UCF-Crime data set, are shown in Tables 6–8 below. The tables report the number of samples, actual positives, actual negatives, true positives, true negatives, and false negatives for each of the categories in the UCF-Crime data set.

**Table 6.** Video anomaly detection confusion matrix reference for FBVAD-*k*NN algorithm.

| Data Set | Samples | Actual Positives | Actual Negatives | True Positives | True Negatives | False Negatives |
|---|---|---|---|---|---|---|
| Abuse | 865 | 194 | 671 | 184 | 671 | 10 |
| Assault | 2523 | 627 | 1896 | 573 | 1896 | 54 |
| Burglary | 1698 | 422 | 1276 | 365 | 1276 | 57 |

**Table 6.** *Cont.*

| Data Set | Samples | Actual Positives | Actual Negatives | True Positives | True Negatives | False Negatives |
|----------|---------|------------------|------------------|----------------|----------------|-----------------|
| Explosion | 757 | 161 | 596 | 156 | 596 | 5 |
| Fighting | 944 | 215 | 729 | 185 | 729 | 30 |
| Road accidents | 347 | 69 | 278 | 63 | 278 | 6 |

The confusion matrix reference and prediction values for FBVAD using the SVM algorithm, based on the UCF-Crime data set, are shown in Table 7 below.

**Table 7.** Video anomaly detection confusion matrix reference for FBVAD-SVM algorithm.

| Data Set | Samples | Actual Positives | Actual Negatives | True Positives | False Negatives |
|----------|---------|------------------|------------------|----------------|-----------------|
| Abuse | 865 | 865 | 0 | 671 | 194 |
| Assault | 2523 | 2523 | 0 | 1896 | 627 |
| Burglary | 1698 | 1698 | 0 | 1276 | 422 |
| Explosion | 757 | 757 | 0 | 596 | 161 |
| Fighting | 944 | 944 | 0 | 729 | 215 |
| Road accidents | 347 | 347 | 0 | 278 | 69 |

The confusion matrix reference and prediction values for PBVAD using the MIM algorithm, based on the UCF-Crime data set, are shown in Table 8 below.

**Table 8.** Video anomaly detection confusion matrix reference for PBVAD-MIM algorithm.

| Data Set | Samples | Actual Positives | Actual Negatives | True Positives | True Negatives | False Negatives |
|----------|---------|------------------|------------------|----------------|----------------|-----------------|
| Abuse | 865 | 31 | 834 | 15 | 834 | 16 |
| Assault | 2523 | 67 | 2456 | 49 | 2456 | 18 |
| Burglary | 1698 | 31 | 1667 | 16 | 1667 | 15 |
| Explosion | 757 | 26 | 731 | 12 | 731 | 14 |
| Fighting | 944 | 33 | 911 | 19 | 911 | 14 |
| Road accidents | 347 | 12 | 335 | 11 | 335 | 1 |

The feature extraction and anomaly detection durations recorded in our tests for each algorithm are shown in Table 9.

**Table 9.** Feature extraction and anomaly detection durations for tested algorithms.

| Data Set | Frame Count | Video Length (min:s) | FBVAD -*k*NN (min:s) | FBVAD -SVM (min:s) | PBVAD -MIM (min:s) |
|----------|-------------|----------------------|----------------------|--------------------|--------------------|
| Abuse | 865 | 00:28 | 00:40 | 00:40 | 00:40 |
| Assault | 2523 | 01:24 | 02:12 | 02:12 | 02:12 |
| Burglary | 1698 | 00:56 | 01:05 | 01:05 | 00:43 |
| Explosion | 757 | 00:25 | 00:36 | 00:36 | 00:29 |
| Fighting | 944 | 00:31 | 00:44 | 00:44 | 00:43 |
| Road accidents | 347 | 00:11 | 00:13 | 00:13 | 00:17 |

Examples of normal and anomalous video frames based on the UCF-Crime data set derived from our tests are shown in Table 10 below.

**Table 10.** Anomalous and normal examples from video frames used in the test study.

| Data Set | Normality Frame | Anomaly Frame-I | Anomaly Frame-II |
|---|---|---|---|
| | Normality<br>Frame No.: 16–130 | Anomaly/Burglary<br>Frame No.: 232 | Anomaly/Burglary<br>Frame No.: 510 |
| Sample-1:<br>Burglary018 |  |  |  |
| | Normality<br>Frame No.: 1–150 | Anomaly/Abuse<br>Frame No.: 420 | Anomaly/Abuse<br>Frame No.: 630 |
| Sample-2:<br>Abuse002 |  |  |  |
| | Normality<br>Frame No.: 1–220 | Anomaly/Assault<br>Frame No.: 450 | Anomaly/Assault<br>Frame No.: 1770 |
| Sample-3:<br>Assault025 |  |  |  |
| | Normality<br>Frame No.: 1–220 | Anomaly/Road Accident<br>Frame No.: 270 | Anomaly/Road Accident<br>Frame No.: 330 |
| Sample-4: Road<br>Accidents002 |  |  |  |

The AUC–ROC graphs obtained from our tests on the abuse, assault, burglary, explosion, fighting, and road accident subdata sets of the UCF-Crime data set are shown in Figure 6 below.

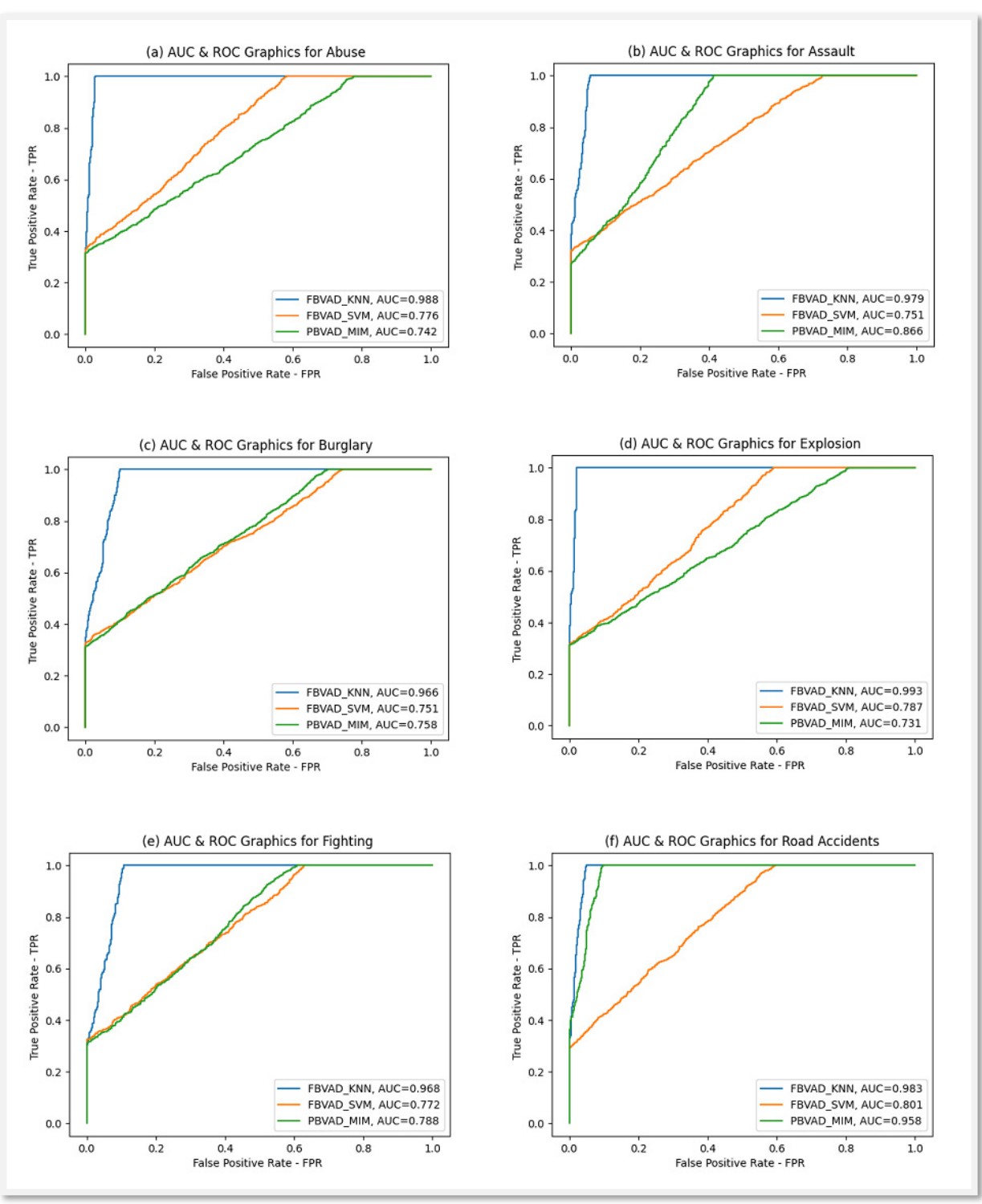

**Figure 6.** AUC–ROC graphs for anomaly detection in the UCF-Crime data set. The AUC–ROC graphs were derived from separate studies on the subdata sets for the following categories: (**a**) abuse; (**b**) assault; (**c**) burglary; (**d**) explosion; (**e**) fighting; and (**f**) road accident.

### *4.4. Comparison of the Results*

The video anomaly detection performance based on the UCF-Crime data set used in our study was also compared with that in other studies, and the results are shown in Table 11 and Figure 7. Compared to two of the previous studies [9,10], our study achieved better results in all categories.

**Table 11.** Comparative accuracy rates for video anomaly detection on UCF-Crime data set.

| Data Set | Sultani [9] | Tian [10] | PBVAD-MIM (Ours) | FBVAD-*k*NN (Ours) |
|---|---|---|---|---|
| Abuse | 70.3 | 55.9 | 74.2 | 98.80 |
| Assault | 54.6 | 70.7 | 86.6 | 97.90 |
| Burglary | 70.1 | 59.5 | 75.8 | 96.60 |
| Explosion | 48.7 | 45.2 | 73.1 | 99.30 |
| Fighting | 70.3 | 70 | 78.8 | 96.80 |
| Road accidents | 59.8 | 55.9 | 95.8 | 98.30 |

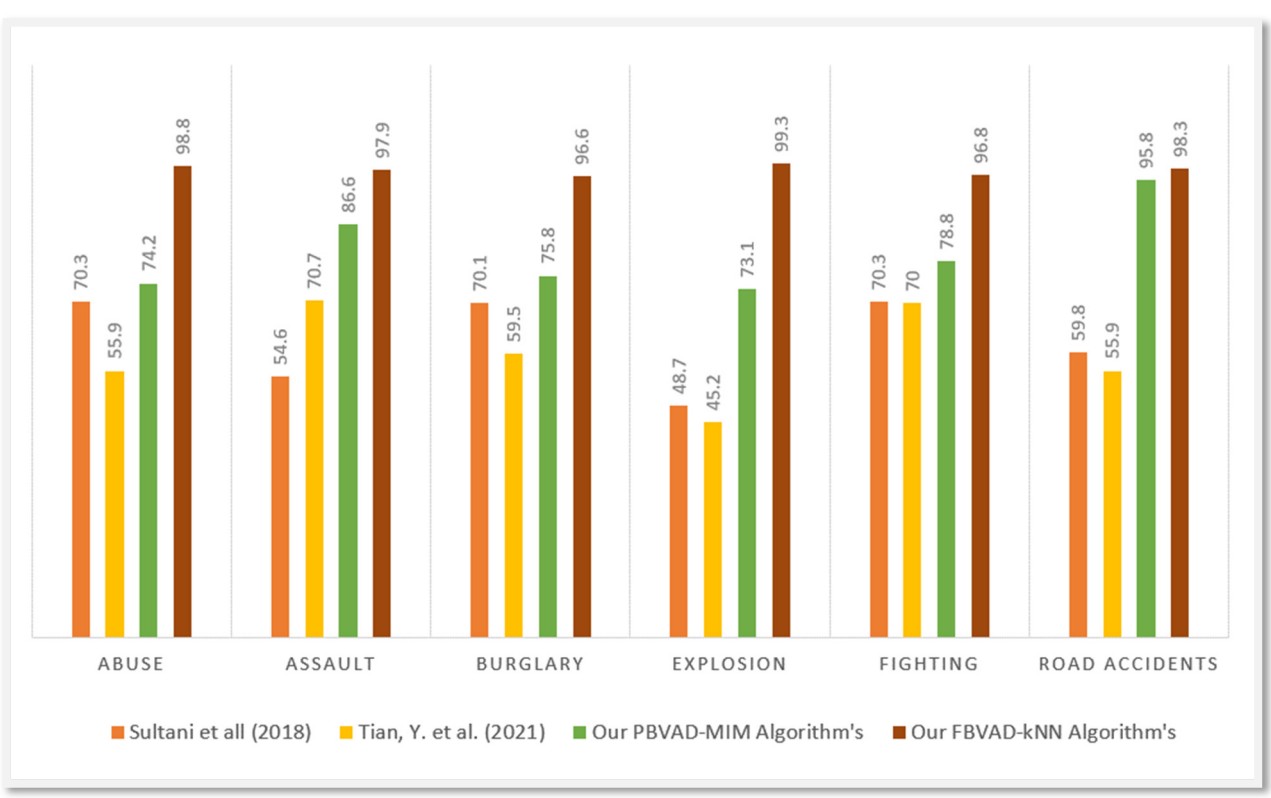

**Figure 7.** Comparison of the accuracy results obtained in our video anomaly detection study with those of others on the UCF-Crime data set [9,10].

### *4.5. Discussion*

In this study, when using frame-based algorithms, the anomaly detection rates for the different categories were similar; however, when using the pixel-based algorithm, large differences were observed between the categories. In pixel-based anomaly detection, the performance rates were better when using video frames with large differences between the pixels (e.g., those in the assault, fighting, and road accident categories) than in video frames with small differences between the pixels (e.g., those in the abuse, burglary, and explosion categories). Through a careful examination of the result tables (Tables 2 and 11)

and graphs (Figures 5 and 7), the category-based differences in the performance rates and the similarities between frame- and pixel-based algorithms can be more clearly understood.

The video anomaly detection performance based on the UCF-Crime data set used in our study was also compared with that of the previous studies [9,10]; our study achieved better results in all categories as shown in Table 11 and Figure 7. In particular, our frame-based video anomaly detection (FBVAD) results derived using the *k*NN algorithm were much better than the results derived using the PBVAD-MIM algorithm, as well as the results of the other studies [9,10].

On the other hand, the accuracy of our results for pixel-based video anomaly detection (PBVAD), derived using the MIM algorithm and applied to the road accident, assault and explosion categories, was higher than that in the other mentioned studies. Furthermore, although the accuracy of our results for PBVAD using the MIM algorithm were better than those of other studies for the assault, burglary, explosion, and fighting categories, the results were similar.

## 5. Conclusions

For this study, the data sets, feature extraction techniques, developed models, algorithms, development environments, analysis results, and accuracy rates reported in studies on similar topics were examined in detail. Subsequently, we proposed a hybrid video anomaly detection model combining multiple machine learning algorithms with pixel-based video anomaly detection (PBVAD) and frame-based video anomaly detection (FBVAD) models. For the PBVAD model, the motion influence map (MIM) algorithm based on spatio–temporal (ST) features was used, while in the FBVAD model, the *k*-nearest neighbors (*k*NN) and support vector machine (SVM) machine learning algorithms were used in a hybrid manner. On the considered UCF-Crime data set, the AUC scores derived using our FBVAD-*k*NN algorithm were as follows: abuse, 98.8%; assault, 97.9%; burglary, 96.6%; explosion, 99.3%; fighting, 96.8%; and road accidents, 98.3%. Furthermore, the success rates when using our PBVAD-MIM algorithm were as follows: abuse, 74.2%; assault, 86.6%; burglary, 75.8%; explosion, 73.1%; fighting, 78.8%; and road accidents, 95.8%.

It was observed that the success rates reported in previous studies based on the UCF-Crime data set using the multiple instance learning (MIL) and robust temporal feature magnitude learning (RTFM) techniques were lower than those in our study. However, our frame-based video anomaly detection (FBVAD) results derived using the *k*NN algorithm were much better than the results derived using the PBVAD-MIM algorithm.

Based on the results of our study, it was observed that anomalous video frames of events such as abuse, assault, burglary, fighting, explosions, and road accidents can be automatically detected in near real time. Thus, the use of the proposed model can make it easier for security guards, operators, or other concerned parties to detect issues faster and take action when using such videos. By detecting anomalous situations in near real time, possible harm can be completely prevented or minimized, providing significant benefits for human health and public safety. At the same time, this study offers many indirect contributions, such as environmental protection and the efficient use of time, facilities, and resources. Another contribution of our study is that its findings can serve as a reference for developing alarm and warning systems based on the detection of anomalous situations in real time, which can serve to increase city and public safety. In the same manner, alarm systems can be developed regarding contexts such as work accidents, burglary, and arson.

Further studies on certain topics may be useful. First, in addition to assessing the performances yielded in similar studies, research should focus on shortening the processing times and reducing energy consumption. Second, new algorithms and techniques which allow for video anomaly detection to be performed more efficiently on less powerful hardware should be developed.

**Author Contributions:** Conceptualization, H.T.; methodology, H.T.; validation, H.T. and M.Z.; formal analysis, H.T.; investigation, H.T.; writing—original draft preparation, H.T.; writing—review and editing, H.T., A.G.; visualization, H.T.; supervision, A.G.; project administration, H.T., A.G., M.Z. and Z.A. All authors have read and agreed to the published version of the manuscript.

**Funding:** The author received no financial support for the research, authorship, and/or publication of this article.

**Data Availability Statement:** The UCF-Crime data set is available in the public domain through the following link: https://www.crcv.ucf.edu/projects/real-world/ (accessed on 1 October 2023).

**Conflicts of Interest:** The author certifies that she has no conflicts of interest in the subject matter or materials discussed in this manuscript. The author certifies that the article is her original work. The article has not received prior publication and is not under consideration for publication elsewhere.

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
