# Peer review of "A Hybrid Approach to Improve the Video Anomaly Detection Performance of Pixel- and Frame-Based Techniques Using Machine Learning Algorithms"

_computation, doi:10.3390/computation12020019_

Round 1

Reviewer 1 Report

Comments and Suggestions for Authors

This study aims to develop an effective anomaly detection model by combining multiple algorithms and techniques. The proposed model is designed to detect anomaly in the combining of frame-level and pixel-level, which are not completely explored in previously research. The paper is well-organized. I suggest the authors further improve this paper before publish from several aspects. 

The literature review is a little disorganized, the author should re-organize line 100~153.

The proposed MIM algorithm is well presented, but only the tabule score results are provided. The author should also show some visual results. For example, when the videos shown in table 5 are input, what about the frame-level or pixel-level anomaly detection results looking like?

Comments on the Quality of English Language

The English used in the paper is clear and understandable.

Author Response

Dear Referee,

Firstly, thank you very much for taking the time to review this manuscript. 

We made the necessary changes in our manuscript, taking your opinions and suggestions into consideration.

In summarize;

  • Our manuscript has been improved by making a general comprehensive revision in its content.
  • The content of the literature review section was developed and subheadings.
  • Anomaly detection results for all algorithms, frame-based and pixel-based, have been added into the manuscript.

Moreover, you can find detailed information about the changes made in our manuscript in the attached PDF file named "MDPI-computation-2793400_Author-response-to-Reviewers_R1".

During this revision process, we colored the manuscript content to make it easier to understand the changes made. The meanings of the colors are as explained below;
Blue: Text and tables that are newly added to the manuscript or whose content has been completely changed are marked in blue.
Green: Expression changes made to the existing text are marked in green.

We upload the new version of our manuscript to the system with the file named "manuscript_computation-2793400_H.TUTAR_V3.2". (Maybe a new name can be given by the system.)

If any other information is needed, we would be grateful if you could inform us. We will get back to you with an answer as soon as possible.

Thank you very much again for your interest and support.

Kind regards
Hayati TUTAR

Reviewer 2 Report

Comments and Suggestions for Authors

1. For writing, the related work section should be improved, not just list some citations

2. In the experimental section, more compared methods should be added. Also, the computation times of the whole system should be reported.

3. In the method section, some equations should be added.

Comments on the Quality of English Language

Some grammar errors and typos:

'In our study proposes a hybrid video anomaly detection model by combining multiple machine learning algorithms with pixel-based video anomaly detection'

Author Response

Dear Referee,

Firstly, thank you very much for taking the time to review this manuscript.

We made the necessary changes in our manuscript, taking your opinions and suggestions into consideration.

In summarize;

  1. The content of the literature review section was developed and subheadings.
  2. Anomaly detection results for all algorithms, frame-based and pixel-based, have been added to the article.
  3. In the section of 3.3.2, k-Nearest Neighbors (kNN) algorithm definition has been added into the content.

With all of these, our manuscript has been improved by making a general comprehensive revision in its content.
Moreover, you can find detailed information about the changes made in our manuscript in the attached PDF file named "MDPI-computation-2793400_Author-response-to-Reviewers_R1".

During this revision process, we colored the manuscript content to make it easier to understand the changes made. The meanings of the colors are as explained below;
Blue: Text and tables that are newly added to the manuscript or whose content has been completely changed are marked in blue.
Green: Expression changes made to the existing text are marked in green.

We upload the new version of our manuscript to the system with the file named "manuscript_computation-2793400_H.TUTAR_V3.2". (Maybe a new name can be given by the system.)

If any other information is needed, we would be grateful if you could inform us. We will get back to you with an answer as soon as possible.

Thank you very much again for your interest and support.

Kind regards
Hayati TUTAR

Reviewer 3 Report

Comments and Suggestions for Authors

Respected Authors,

The authors have proposed "hybrid video anomaly detection model by combining multiple machine learning algorithms with pixel-based video anomaly detection (PBVAD) and frame-based video anomaly 17 detection (FBVAD) models." However, there is few changes needs to be done before publication.

1. The references can be found with reference to particular methods and techniques at pages 103 and 104. It was extremely unsettling in its current state say "video event anomaly detection 102 [16,23,29,54,66,76,78,84] and video anomaly detection in crowded areas 103 [15,19,26,30,53,55,59]"

2. Misspelled as "Hidden Markove Model (HMM)" update it throughout the paper.

3. A thorough and well-reasoned discussion of the proposed system is required in Section 4.4, Comparison of the Results and Discussion. Please add one more paragraph if you can. 

Comments on the Quality of English Language

Nil

Author Response

Dear Referee,

Firstly, thank you very much for your suggestions and evaluations about our manuscript.

We made the necessary changes in our manuscript, taking your opinions and suggestions into consideration.

In summarize;

  1. The content of the literature review section was developed and subheadings.
  2. In order to make our manuscript more clear, we excluded the Hidden Markov Model - HMM algorithm, which had low test results, from the scope of our study.
  3. We have added new section with name is "4.5. Discussion" for discussion of the our proposed system. And we revised its content.

With all of these, our manuscript has been improved by making a general comprehensive revision in its content.
Moreover, you can find detailed information about the changes made in our manuscript in the attached PDF file named "MDPI-computation-2793400_Author-response-to-Referees_R1".

During this revision process, we colored the manuscript content to make it easier to understand the changes made. The meanings of the colors are as explained below;
Blue: Text and tables that are newly added to the manuscript or whose content has been completely changed are marked in blue.
Green: Expression changes made to the existing text are marked in green.

We upload the new version of our manuscript to the system with the file named "manuscript_computation-2793400_H.TUTAR_V3.2". (Maybe a new name can be given by the system.)

If any other information is needed, we would be grateful if you could inform us. We will get back to you with an answer as soon as possible.

Thank you very much again for your interest and support.

Kind regards
Hayati TUTAR

Round 2

Reviewer 2 Report

Comments and Suggestions for Authors

I have no further questions